# Di-Tyrosine Crosslinking and *NOX4* Expression as Oxidative Pathological Markers in the Lungs of Patients with Idiopathic Pulmonary Fibrosis

**DOI:** 10.3390/antiox10111833

**Published:** 2021-11-18

**Authors:** Sanja Blaskovic, Yves Donati, Isabelle Ruchonnet-Metrailler, Tamara Seredenina, Karl-Heinz Krause, Jean-Claude Pache, Dan Adler, Constance Barazzone-Argiroffo, Vincent Jaquet

**Affiliations:** 1Department of Pediatrics, Gynecology and Obstetrics, Children’s Hospital, 1211 Geneva, Switzerland; Sanja.Blaskovic@unige.ch (S.B.); Yves.Donati@unige.ch (Y.D.); Isabelle.Ruchonnet-Metrailler@hcuge.ch (I.R.-M.); 2Department of Pathology and Immunology, Medical School, University of Geneva, 1211 Geneva, Switzerland; tamara.seredenina@gmail.com (T.S.); Karl-Heinz.Krause@unige.ch (K.-H.K.); Vincent.Jaquet@unige.ch (V.J.); 3Service of Clinical Pathology, Department of Pathology and Immunology, Medical School, University of Geneva, 1211 Geneva, Switzerland; Jean-Claude.Pache@hcuge.ch; 4Division of Pulmonary Diseases, Department of Medicine, Geneva University Hospitals, 1211 Geneva, Switzerland; Dan.Adler@hcuge.ch; 5READS Unit, Medical School, University of Geneva, 1211 Geneva, Switzerland

**Keywords:** idiopathic pulmonary fibrosis, NADPH oxidase, *NOX4*, di-tyrosine, immunohistochemistry, human lung tissue

## Abstract

Idiopathic pulmonary fibrosis (IPF) is a noninflammatory progressive lung disease. Oxidative damage is a hallmark of IPF, but the sources and consequences of oxidant generation in the lungs are unclear. In this study, we addressed the link between the H_2_O_2_-generating enzyme NADPH oxidase 4 (*NOX4*) and di-tyrosine (DT), an oxidative post-translational modification in IPF lungs. We performed immunohistochemical staining for DT and *NOX4* in pulmonary tissue from patients with IPF and controls using validated antibodies. In the healthy lung, DT showed little or no staining and *NOX4* was mostly present in normal vascular endothelium. On the other hand, both markers were detected in several cell types in the IPF patients, including vascular smooth muscle cells and epithelium (bronchial cells and epithelial cells type II). The link between *NOX4* and DT was addressed in human fibroblasts deficient for *NOX4* activity (mutation in the *CYBA* gene). Induction of *NOX4* by Transforming growth factor beta 1 (TGFβ1) in fibroblasts led to moderate DT staining after the addition of a heme-containing peroxidase in control cells but not in the fibroblasts deficient for *NOX4* activity. Our data indicate that DT is a histological marker of IPF and that *NOX4* can generate a sufficient amount of H_2_O_2_ for DT formation in vitro.

## 1. Introduction

Idiopathic pulmonary fibrosis (IPF) is a progressive interstitial lung disease leading to lung tissue stiffness and fibrosis, decreased lung function and eventually respiratory failure. It is estimated that three million people are affected by IPF, worldwide. The median survival of IPF is estimated at 2–3 years after diagnosis [1,2]. IPF is characterized by the aberrant proliferation and activation of fibroblasts as well as the presence of myofibroblasts that secrete excessive collagen and extracellular matrix (ECM) leading to the scarring of lung tissue and destruction of the alveolar epithelium (reviewed in [3]). Lung remodeling is thought to occur following repetitive injuries leading to alveolar epithelial cells (AEC) apoptosis, proliferation of fibroblasts and Transforming growth factor beta 1 (TGFβ1)-induced myofibroblast differentiation as well as endothelial dysfunction. However, knowledge of the underlying molecular mechanisms leading to IPF is scarce and the clinically approved antifibrotic agents Nintedanib and Pirfenidone provide only limited benefit and are associated with adverse effects. Characterization of novel alternative pathogenic pathways is strongly needed to improve our knowledge on this devastating disease and develop novel classes of therapeutics.

A key hallmark of IPF pathogenesis is the presence of high levels of reactive oxygen species (ROS) in the lungs (reviewed in [4]). Although, the sources of ROS can be multiple, *NOX4* represents a key enzymatic source of ROS in IPF lungs and has been associated to fibrogenic properties in the lungs [5,6]. *NOX4* is a member of the family of nicotinamide adenine dinucleotide phosphate (NADPH) oxidases, whose sole function is the generation of ROS [7]. *NOX4* expression is strongly upregulated in fibroblasts from IPF patients and following treatment with the fibrogenic cytokine TGFβ1 leading to extracellular H_2_O_2_ production [5]. In addition, *NOX4*-deficient mice show decreased pulmonary fibrosis and AEC cell death in a mouse model of bleomycin-induced lung fibrosis [6]. *NOX4* inhibitors are currently investigated in clinical trials for IPF (NCT03865927). Nevertheless, the precise molecular mechanisms by which *NOX4*-derived ROS contribute to IPF pathology are not known. ROS are often considered as harmful molecules directly damaging cellular components, but ROS (and in particular H_2_O_2_) also exert a physiological function as signaling molecules [8]. In fact, H_2_O_2_ can serve as a substrate for enzymatic reaction catalyzed by peroxidases in various biological reactions, such as the generation of hypochlorous acid (HOCl) for germ killing by granulocytes and thyroid hormone synthesis [9]. One such specific reaction is called di-tyrosine (DT) crosslinking. In vitro, DT formation is catalyzed by a heme-containing peroxidase in the presence of H_2_O_2_ leading to the formation of a covalent carbon−carbon bond between two tyrosine residues of the same, or two adjacent proteins [10]. This phenomenon has physiological relevance and is well described in invertebrates. For instance, during egg fertilization of the sea urchin egg, the combined activity of a NADPH oxidase (Udx1) and a secreted ovoperoxidase is required for crosslinking of protein tyrosyl residues to create a stiffened ECM around the egg to avoid polyspermy [11]. However, DT is also present in different pathological conditions such as Alzheimer’s disease [12], Parkinson’s disease [13], atherosclerotic plaque formation [14], and IPF. A study by Pennathur and collaborators used mass spectrometric quantification of oxidative tyrosine modifications and documented a significant increase in the plasma of patients with interstitial lung disease and in the lungs of bleomycin-treated mice [15]. As of today, we are still unaware whether DT is formed in the lungs of IPF patients, and how its levels and localization contribute to IPF pathogenesis. A recent study has shown that the DT cross-linking of fibronectin, a key protein of the ECM, alters functional fibronectin characteristics in vitro [16]. In fact, ECM properties and glycogen crosslinking are known to be altered in IPF patients, thereby contributing to tissue stiffness [17]. In terms of mechanism, neither the H_2_O_2_ sources nor the potential peroxidases involved in DT formation in humans are known.

Thus, we aimed at addressing the localization and levels of DT as well as *NOX4* as the potential source of H_2_O_2_ involved in DT generation in the lungs of patients with IPF. We showed that normal lungs were virtually devoid of DT while lung tissue from IPF patients were characterized by a significant DT staining. *NOX4* was a sufficient source of H_2_O_2_ to form DT in vitro, but *NOX4* expression in IPF tissue did not clearly correlate with DT localization, especially in the smooth muscle cells of IPF lungs. Altogether, we conclude that DT and *NOX4* expression patterns are strongly affected in IPF while DT represents a key novel histopathological feature of IPF.

## 2. Materials and Methods

### 2.1. Antibodies, Enzymes and Reagents

The anti-di-tyrosine monoclonal antibody clone 1C3 was purchased from JaiCa, Fukuroi, Japan and the rabbit monoclonal antibody directed against the C-terminus of *NOX4* was a kind gift of Prof Jansen Dürr, University of Innsbruck (Innsbruck, Austria). Isotype negative control antibodies were the following: mouse IgG2a (DAKO, X0943) for DT and recombinant rabbit IgG, monoclonal isotype control [EPR25A] (ACAM, ab172730) for *NOX4*. DAKO REAL Detection system peroxidase/DAB+, rabbit/mouse, K5001 was used for IHC on human tissues. Transforming Growth Factor-β1 human was from Sigma (T7039, Saint Louis, MO 63103, USA). Alexa Fluor 488 tyramide Reagent (TSA) was from Life Technology, (B40953, Eugene, Oregon, USA). Fetal calf serum was purchased from Chemie Brunschwig (K007310G). Fluorescent secondary antibody for DT immunofluorescence was goat anti-mouse IgG DyLight594 (Jackson ImmunoResearch, Cambridge, UK, 115-515-003). Panexin basic was purchased from Pan Biotech (P04-96090, Aidenbach, Germany). Phosphate-Buffered Saline, PBS (14190, Life Technologies, Paisley, Scotland, UK), Hepes 1M (15630, Life Technologies, Paisley, Scotland, UK) and Hanks’ Balanced Salt Solution (HBSS), calcium, magnesium, HBSS++ (14025, Life Technologies, Paisley, Scotland, UK) were from GIBCO. L-tyrosine was from Carl Roth (1741.1, Karlsruhe, Germany). Normal goat serum (NGS) was from Invitrogen (10000C, Waltham, Massachusetts, USA), DAPI from Roche Diagnostics (10236276001, Basel, Switzerland), FluorSave Reagent (345789) and H_2_O_2_ (107209) were from Merck (Merck KGaA, Darmstadt, Germany).

### 2.2. Gene Expression in Fibroblasts

The human lung fibroblast cell line MRC5 (ATCC CCL-171) and primary human skin fibroblasts isolated from a healthy donor (hSFs) and a patient carrying mutation in *CYBA* gene (hSFp22 deficient) [18], were maintained in Dulbecco’s modified Eagle’s medium (DMEM) supplemented with fetal bovine serum (FBS, 10%), penicillin (100 U/mL), and streptomycin (100 µg/mL) at 37 °C in air with 5% CO_2_. For differentiation experiments, the cells were kept in a serum-free medium for 24 h and treated with 2 ng TGF-β1 for 24 h and collected for RNA extraction.

RNA was extracted using RNeasy mini kit (Qiagen, Dusseldorf, Germany) according to manufacturer’s protocol. Five hundred nanograms (500 ng) were used for cDNA synthesis using the PrimeScript RT reagent kit (Takara, Saint-Germain-en-Laye, France) following the manufacturer’s instructions. Real-time PCR was performed using the SYBR green assay at the Genomics Platform, National Center of Competence in Research Frontiers in Genetics (Geneva, Switzerland), on a 7900HT SDS system (Applied Biosystems, Foster City, USA). The efficiency of each primer was assessed with serial dilutions of cDNA. Relative expression levels were calculated by normalization to the geometric mean of two housekeeping genes, β2-microglobulin and *GAPDH*, as previously described [19]. Normalized quantities are reported as mean ± SEM. The sequences of the primers used in this study are documented in Table 1.

### 2.3. TSA Assay and DT Immunofluorescence

Fibroblasts were cultured on glass coverslips as described above. A starvation step was performed by adding new media with low FCS concentration (0.01% for HSF cells) or culture media supplemented with 10% panexin, and 0.1% FCS for MRC5 cells for 24 h. TGFβ1 was added in the starvation media (5 ng/mL) for 7 days with medium change every 2–3 days. On the day of the experiment a 100× dilution of TSA was added with or without HRP (5 U/mL) and with or without H_2_O_2_ (10 µM) in TRIO buffer (1 mM Hepes, 1 mM L-Tyrosine in HBSS++) for 1 h at RT. Cells were washed 2× with HBSS++ and fixed with 4% paraformaldehyde for 15 min at room temperature (RT) and washed with PBS. Cells were blocked for 1 h at RT in blocking solution (PBS, 5% NGS, 0.3% TritonX-100). Cells were incubated with primary di-tyrosine antibody (dilution 1:400) at RT. After 1 h, cells were washed 3 times for 5 min with PBS at RT and a secondary anti-mouse antibody conjugated to D594 was added for 1 h (dilution 1:200). Cells were again washed 3 times for 5 min at RT and cell nuclei were stained with DAPI (1 µg/mL in PBS) 5 min at RT. Coverslips with cells were mounted with FluorSave and imaged with Axiocam Fluo (Carl Zeiss Microscopy GmbH, Jena, Germany).

### 2.4. Human Lung Samples

Human lung tissue samples used in this study (Table 2) were obtained from the department of Pathology, University Geneva hospital. Samples were collected either on biopsies or on necropsies. Clinical description was limited to definitive diagnosis made by the pathologist, the gender and the age of the patient. Samples were irreversibly anonymized, dated, processed and analyzed according to the Swiss Medical ethical guidelines and recommendations (Senate of the Swiss Academy of Medical Sciences, Basel, Switzerland, 23 May 2006). The study was approved by the regional research committee of the University Geneva Hospital (NAC10-052R; NAC 11-027R).

### 2.5. Immunohistochemical Staining

Paraffin-embedded sections (5 µm) of human control and IPF affected lungs were stained according to the cell signaling technology standard protocol [20] using DAKO Envision kit (K4011, Agilent Technologies, CA, USA) using DT and *NOX4* antibodies.

Briefly, paraffin-embedded samples were deparaffinized using xylene and 95–100% ethanol and subsequently hydrated in H_2_O.

*NOX4* labeling required pressure and heat-induced epitope retrieval (20 bar) in Tris–EDTA, pH 9.0 (10 mM/1 mM) buffer. DT did not require antigen retrieval. Endogenous peroxidases were blocked with DAKO peroxidase block solution. Both primary and secondary antibodies were diluted with DAKO antibody diluent. The anti-DT mouse monoclonal antibody and its mouse IgG2a isotype control were used at 0.25 µg/mL. The anti-*NOX4* rabbit monoclonal antibody and its rabbit monoclonal Ig isotype control were used at 2.5 µg/mL). We applied the primary antibodies for 1 h at RT. Finally, labeled polymer-horseradish peroxidase (HRP) anti-rabbit or anti-mouse (DAKO Envision system, Agilent Technologies, CA, USA) was used for 30 min at RT and the signal was visualized with diaminobenzidine (Envision system, Dako SA, Geneva, Switzerland). Sections were counterstained with hematoxylin (BioGnost, Zagreb, Croatia). Images were acquired using Axioscan Z1 microscope (Carl Zeiss Microscopy GmbH, Jena, Germany) and analyzed using the Definiens Developer XD™ 2.7 software (Definiens AG, Munich, Germany). Quantification of DT and *NOX4* was done in ImageJ (version 1.51, NIH, Rockville Pike, Bethesda, Maryland) on randomly selected 10 images/sample and statistical analysis was carried out with GraphPad PRISM (Prism 8.0.2, GraphPad Software, San Diego, CA, USA) using the Mann−Whitney nonparametric test. At least 10 adjacent sections were stained for each antibody. Control tissue consisted of adjacent healthy tissue for patients nos. 6–9 and from postmortem tissue for patient no. 10.

## 3. Results

Our study aimed at evaluating the pattern of expression of the H_2_O_2_-generating enzyme *NOX4* and the presence of oxidative post-translational modification DT in human lungs. The *NOX4* antibody used in this study was a rabbit monoclonal antibody directed against the intracellular C terminus of human *NOX4* produced in the laboratory of Professor Pidder Jansen-Dürr, Institute for Biomedical Ageing Research. University of Innsbruck. This antibody belongs to the small list of available validated *NOX4* antibodies, suitable for Western blot and IHC [21]. We previously documented the specificity of this antibody over other NOX isoforms [22] and its specificity for immunohistochemical detection in human tissue samples [23]. The mouse monoclonal DT antibody was generated using a DT conjugate as immunogen and selected for DT specificity [24]. In order to address the specificity of the DT antibody and the biological significance of *NOX4* in DT formation, we used an in vitro fibroblast experimental system inspired by the methodology described by Larios et al. [25].

### 3.1. Generation and Immunodetection of DT in Fibroblast Cell Culture

We used MRC5 lung fibroblasts treated with TGFβ1 to induce myofibroblast differentiation as evidenced by increased expression of *NOX4*, alpha smooth muscle actin (α-SMA) and the ECM proteins fibronectin and collagen 1a [16] (Figure 1A). The heme-containing peroxidase (HRP) and fluorescein-tyramide, a fluorescently labeled chemical analogue of tyrosine, were added to the fibroblast culture. Upon H_2_O_2_ addition, HRP converted fluorescein-tyramide into a highly-reactive tyramide radical that forms dimers with tyrosine residues allowing the detection of DT crosslinks formed with proteins of the ECM (Figure 1B). After washing unbound fluorescein tyramide, immunofluorescent staining using the anti-DT monoclonal antibody was performed. A striking colocalization of bound fluorescent tyramide and DT staining was observed (Figure 1B) confirming that the antibody recognized the DT formed between fluorescein tyramide and ECM proteins by the coordinated action of H_2_O_2_ and HRP.

### 3.2. NOX4-Derived H_2_O_2_ Is Sufficient for DT Generation in Skin Fibroblasts

In order to address whether *NOX4* was able to generate a sufficient amount of H_2_O_2_ upon activation by TGFβ1 to generate DT, we used primary skin fibroblasts isolated from a patient with a mutation in *CYBA*, the gene coding for p22phox [18], a necessary subunit for *NOX4* activity [26]. Similar to MRC5, *NOX4* was induced in both control and *CYBA*-deficient skin fibroblasts (Appendix A). Interestingly, a weak overlapping staining was detected using both fluorescent tyramide and DT staining in the WT fibroblasts (Figure 1C, 1st and 2nd panel), which was completely absent in *CYBA*-deficient fibroblasts, whose *NOX4* is not active (Figure 1C, 3rd panel). This suggests that *NOX4* generates a sufficient amount of H_2_O_2_ for DT generation in differentiated skin myofibroblasts, provided that a heme containing peroxidase and tyrosine-rich proteins of the ECM are present.

### 3.3. Global DT Staining Is Significantly Increased in the Lungs of IPF Patients

In order to evaluate the presence of DT crosslinking and *NOX4* expression in the lungs of IPF patients, we performed immunohistochemistry (IHC) for both markers. Isotype control antibodies were used to confirm the specificity of the staining (Appendix A). We detected high levels of DT in the lungs of IPF patients, indicative of a wide dissemination of ECM proteins and oxidative events. On the other hand, DT staining was barely above background in control tissue (Figure 2A,B). *NOX4* staining was more localised than DT staining and *NOX4* levels varied widely between individuals. Altogether, no significant change of global *NOX4* levels was detected in the lungs of IPF patients compared to controls (Figure 2C,D).

We further analyzed the lung material used in this study and focused on the cell types stained by *NOX4* and DT antibodies, namely smooth muscle cells, alveolar type II and bronchial epithelial cells and endothelial cells.

### 3.4. DT and NOX4 Are Present in Thickened Vessel Walls in the Lungs of IPF Patients

Vascular remodeling is a key pathological feature in the scarred areas of IPF lungs. It is characterized by the proliferation of pulmonary arterial smooth muscle cells in response to hypoxia leading to the thickening of the media, the smooth muscle layer around the blood vessels. Vascular smooth muscle cells in control lung tissues were negative for DT and *NOX4* (Figure 3A). However, fibrotic regions of IPF lungs showed abnormal vascular architecture and a strong staining for both DT and *NOX4* in pulmonary arterial smooth muscle cells (Figure 3B). At higher magnification, pulmonary arterial smooth muscle cells appeared highly positive for DT and to express high *NOX4* levels (insets in Figure 3B). While DT staining appeared cytoplasmic, *NOX4* staining was mainly found in perinuclear regions of smooth muscle cells. This indicates an upregulation of *NOX4* and DT in a specific vascular pathological modification of IPF and may suggest a potential role of *NOX4* in the generation of H_2_O_2_ needed for DT formation in proliferative pulmonary arterial smooth muscle cells.

### 3.5. DT Staining and NOX4 Expression in Bronchi and Alveolar Epithelial Cells Type II of IPF Patients

The pulmonary alveolar epithelium is essential for the air–blood barrier function of the lungs. It is mainly composed of alveolar epithelial cells type I (AT1) and type II (AECII) cells. AT1 cells are abundant, large squamous cells involved in gas exchange, while AECII cells are smaller, cuboidal cells mostly involved in synthesis and secretion of the lung surfactant. Contrary to endothelial cells, smooth muscle cells, alveolar epithelial cells type II and macrophages which can be recognized by their tissue localization and morphology, it was impossible to correctly identify AT1 cells by IHC and hence we only looked at AECII cells.

AECII were not stained with DT and *NOX4* antibodies in control lung tissue. On the other hand, although the alveoli were severely damaged, both *NOX4* and DT antibodies decorated the remaining AECII cells in IPF (Figure 4B), consistent with a possible role of *NOX4* in DT formation in AECII.

In normal AECII epithelial cells and bronchi, very little staining was observed for both markers (Figure 4A,B, upper panels). However, both AECII and bronchial cells were stained in IPF samples with *NOX4* and DT antibody with a different intensity, depending on the patient (Figure 4A,B, lower panels).

### 3.6. Different Expression of DT and NOX4 in Vascular Endothelium in Normal and IPF Tissue

Interestingly, while IPF tissue globally showed elevated levels of DT, only partial DT staining was observed in pulmonary endothelial cells (EC) (Figure 5). On the other hand, high *NOX4* levels were present in the endothelial cells of large vessels in both control and IPF tissue. *NOX4* staining was also detected in capillary endothelial cells in some samples (data not shown). Due to the high specificity of *NOX4* staining in endothelial cells and the fact that *NOX4* is generating H_2_O_2_ constitutively, the absence of DT staining in two out of four samples indicates that the conditions required for DT formation are not fulfilled in the intimal parts of the vessels.

Altogether, DT staining was found in none of our control samples, but was present in SMC, AEC and AECII in all tested IPF patients (Figure 6A). *NOX4* staining was present in EC in both control and IPF samples, while AECII and SMC cells were mainly stained only in IPF patients (Figure 6B). Note that both antibodies stained macrophages in all samples (Figure 6A,B).

## 4. Discussion

In this study, we showed that DT is a novel specific histopathological marker of IPF that can be detected with a specific antibody in lung tissue. We showed using differentiated myofibroblasts that *NOX4* generated a sufficient amount of H_2_O_2_ for DT formation in vitro and that *NOX4* and DT stained both proliferative pulmonary smooth muscle cells and epithelial bronchial and AECII cells. However, the endothelial cells of blood vessels were stained predominantly with *NOX4* and independently of the disease, though some IPF samples also showed the staining with DT.

Our data confirm that DT accumulation in the lungs is a pathological feature of IPF. Indeed, our data confirmed previous reports showing increased levels of proteins containing DT modifications, in particular the tyrosine-rich ECM protein, fibronectin, in the plasma of patients with fibrotic interstitial lung disease and the bleomycin-induced IPF mouse model [15,16].

Since DT bridges are covalent and most likely irreversible in vivo, a pharmacological strategy using small chemical molecules or anti-DT antibodies is unlikely to be successful. However, the well-described mechanism leading to DT formation provides a rational for future therapies for IPF. Such an approach would aim at inhibiting one or several of the components necessary for DT formation, namely, (i) abundant levels of proteins of the ECM, (ii) one or several sources of H_2_O_2_ and (iii) the presence of an active heme-containing peroxidase [16,24,25,27]. Existing therapeutic approaches in IPF that target the fundamental mechanisms of fibrosis leading to the deposition of ECM, including the proliferation and differentiation of fibroblasts are comprehensively reviewed in [28,29] and will not be developed here.

More than 40 enzymes are described as generating H_2_O_2_ when they are active [8]. Among them, NADPH oxidases are highly relevant sources of H_2_O_2_ in the context of DT formation in IPF lungs. NOX is known to coordinate their activity with specific heme-containing peroxidases for key biological functions in humans and other organisms, including host defense and thyroid hormone synthesis (reviewed in [9]). Several NOX isoforms are expressed in the lungs, but most studies point towards a contribution of *NOX4* in IPF. *NOX4* is upregulated in IPF lungs while *NOX4*-deficient mice are partially protected in a bleomycin mouse model of IPF [5,6]. *NOX4* expression is induced in fibroblasts concomitantly with ECM proteins following TGFβ1 treatment. Interestingly, high DT staining in dermal tissue observed during physiological wound healing is absent in *NOX4*-deficient mice [30]. The fact that p22^phox^-deficient skin fibroblasts were not able to generate DT crosslinking with TSA, in spite of the presence of high levels of ECM and an active heme peroxidase, supports a role of *NOX4* in DT formation at least in the skin. No DT was observed in the sites with the highest *NOX4* levels in endothelial cells in healthy tissue, but we detected some DT staining in two out of four IPF samples. One possibility is that in endothelial cells, *NOX4* may generate H_2_O_2_ in the luminal side where ECM proteins or heme-containing peroxidases are absent. Alternatively, although the absence of colocalization does not exclude the diffusion of *NOX4*-derived H_2_O_2_ to other lung regions, *NOX4*-dependent DT formation might be limited to the fibrotic foci. Other sources of H_2_O_2_ might contribute to DT formation in IPF tissue. The dual oxidases DUOX1 and DUOX2 are expressed in airway epithelia where they generate H_2_O_2_ upon activation [31,32]. In particular DUOX1 and 2 may play a role in host defense in lung tissue through a coordinated function with the heme-containing lactoperoxidase [33]. DUOX1 was shown to be induced following lung injury, while DUOX1-deficient mice have an attenuated fibrotic phenotype in a bleomycin model of IPF [34]. However, to our knowledge, in spite of a strong rationale, the involvement of the DUOX1/lactoperoxidase coupling in DT formation in the lungs has not been studied yet.

## 5. Conclusions

In conclusion, our IHC study of postmortem and biopsy material describes elevated DT and specific DT and *NOX4* expression patterns as typical features of IPF human pathology. The validation of a specific antibody against DT allowing the detection of a novel histopathological marker of IPF pathology grants a great impact in future studies aiming at dissecting the pathogenic roles played by DT formation in IPF. Our study opens a number of important questions for further investigation. including the timing of DT formation in IPF lungs, the types of ECM proteins undergoing DT formation, the alternative sources of H_2_O_2_ and the necessary heme-containing peroxidases leading to high DT in IPF. Exploration of these questions might lead to the discovery of novel therapeutics for IPF and potentially other fibrotic diseases.

## Figures and Tables

**Figure 1 antioxidants-10-01833-f001:**
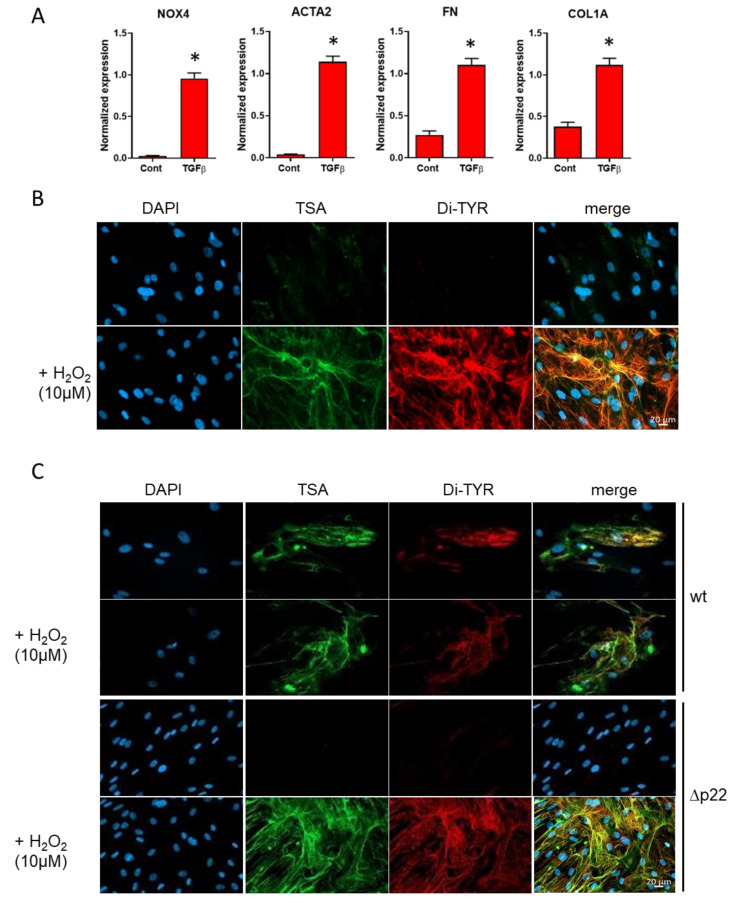
(**A**) TGFβ1-induced upregulation of the expression of *NOX4*, the myofibroblast marker alpha-SMA and proteins of the ECM in MRC5 human lung fibroblasts. *N* = 4, * *p* < 0.05 using Mann−Whitney nonparametric test. (**B**). Fluorescent microscopy of DT in TGF-β1-treated MRC5. Addition of H_2_O_2_ in presence of HRP cells induces crosslinking of the fluorescein tyramide (TSA, green) with the extracellular matrix network. The cross-links are detected by the DT antibody (red) and colocalize with crosslinked TSA (yellow). (**C**). Similar experiment with human skin fibroblasts from a healthy donor and a patient carrying a p22^phox^ mutation. Deletion of p22^phox^ completely abolishes the formation of DT in the absence of exogenous H_2_O_2_. Blue, DAPI staining of cell nuclei. *NOX4*: NADPH oxidase isoform 4; ACTA 2: Actin Alpha2, Smooth Muscle; FN: Fibronectin; COL1A: Collagen1A; DAPI: 4′,6-diamidino-2-phenylindole; TSA: Alexa Fluor 488 tyramide Reagent; Di-TYR: dityrosine; TGFβ: Transforming growth factor β; Cont: control; HRP: horseradish peroxidase.

**Figure 2 antioxidants-10-01833-f002:**
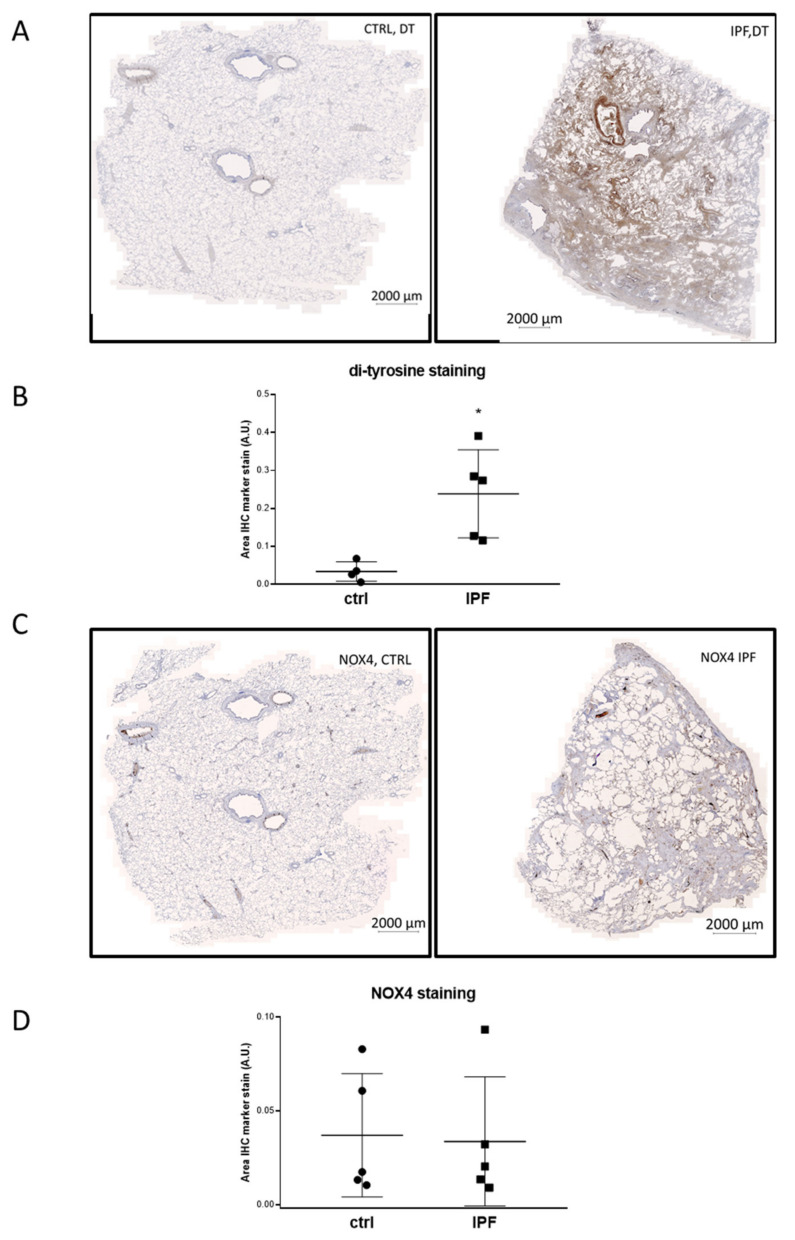
Representative images of DT (**A**) and *NOX4* (**C**) staining in the indicated patients are shown for lungs. Quantification of staining for DT (**B**) and *NOX4* (**D**). * *p* < 0.05 using Mann−Whitney nonparametric test. CTRL: control; IPF: idiopathic pulmonary fibrosis.

**Figure 3 antioxidants-10-01833-f003:**
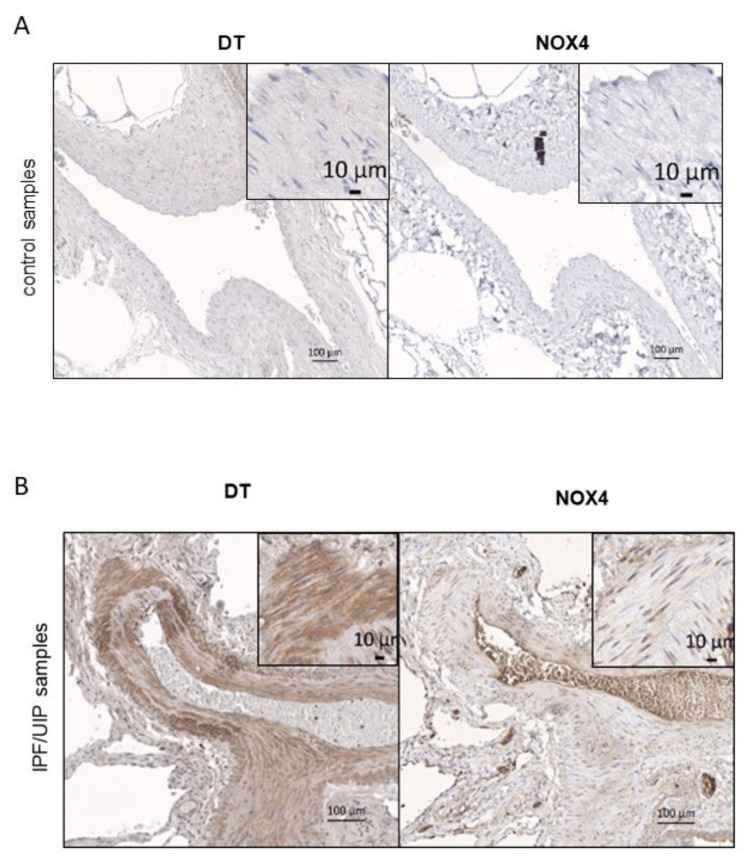
Representative images for DT ad *NOX4* staining of blood vessel smooth muscle cells in the control (**A**) and IPF samples (**B**) of human patients are shown. Inset shows a 10× higher magnification and depicts typical smooth muscle cell staining in these samples. DT and *NOX4* staining are shown in brown.

**Figure 4 antioxidants-10-01833-f004:**
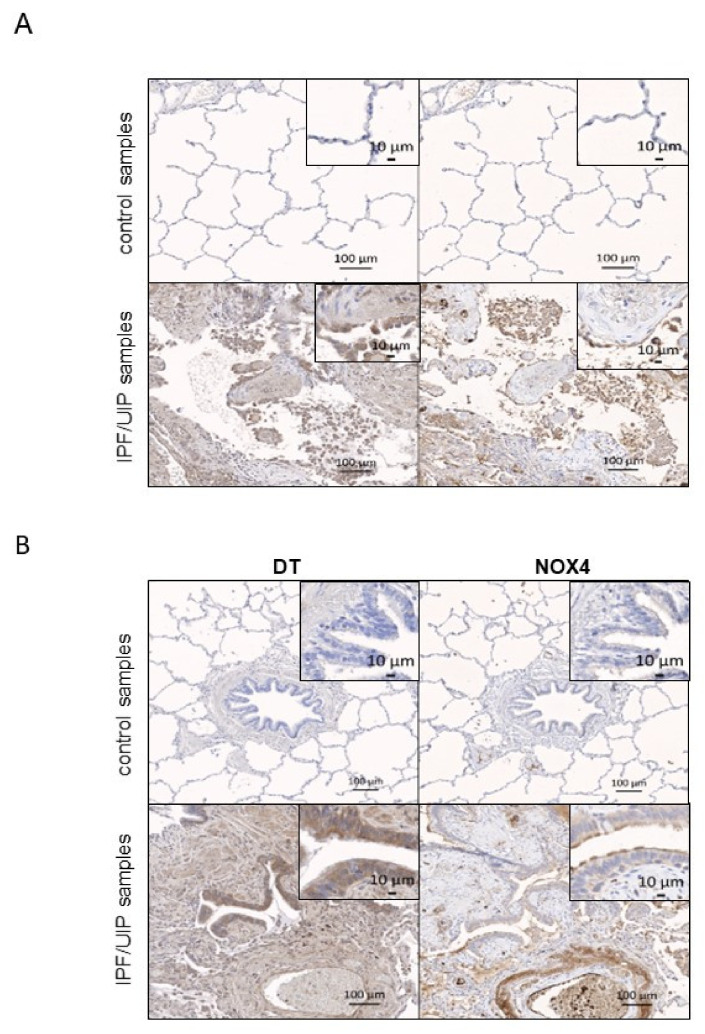
Representative images for DT and *NOX4* staining of AECII cells (**A**) and bronchi (**B**) in the control (upper panel) and IPF/UIP samples (lower panel) of human patients are shown. Insets shows a 10× higher magnification and depicts AECII (**A**) and ciliary bronchial cells (**B**) in these samples. DT and *NOX4* staining are shown in brown.

**Figure 5 antioxidants-10-01833-f005:**
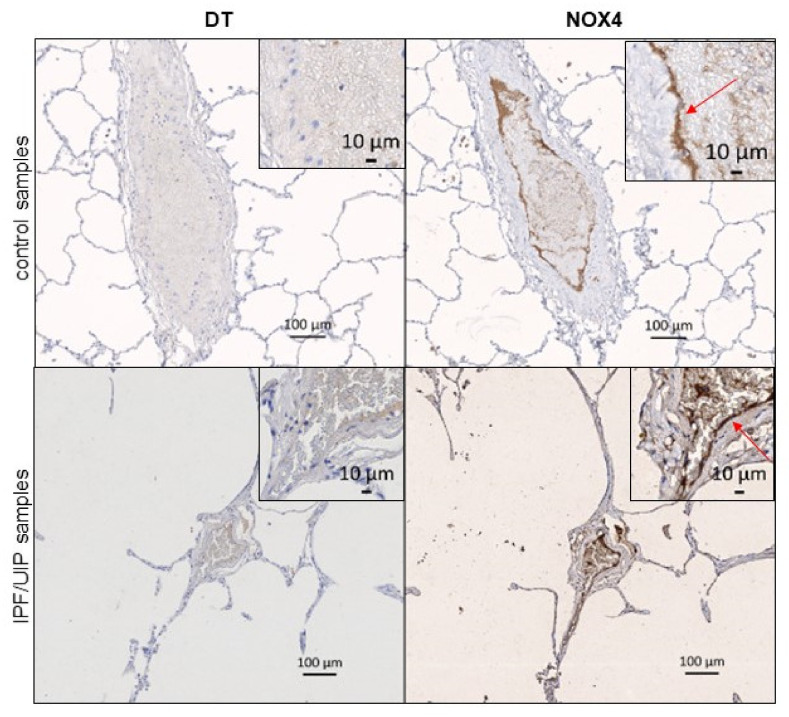
Representative images for DT and *NOX4* staining of blood vessel endothelial cells in the control (upper panels) and the IPF/UIP samples (lower panels) from human patients are shown. Inset shows a 10X higher magnification and depicts typical intimal endothelial region of these samples. DT and *NOX4* staining are shown in brown. The red arrows indicate the *NOX4* endothelial staining.

**Figure 6 antioxidants-10-01833-f006:**
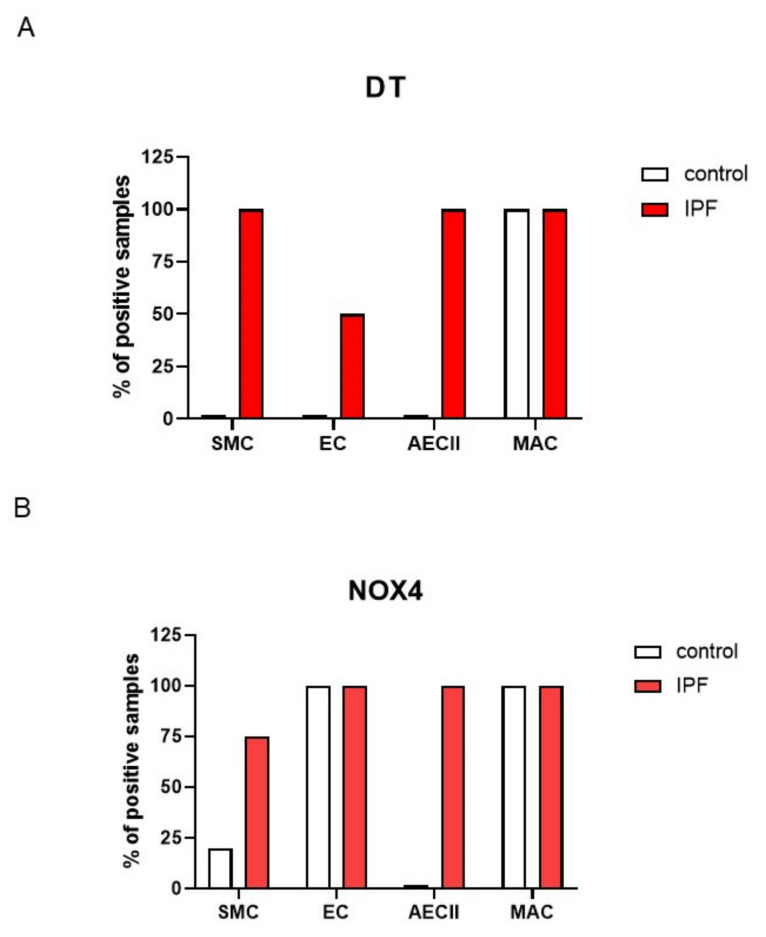
Histograms documenting the number of samples positive for DT and *NOX4* for each cellular subtype. All 4 IPF patients were positive for DT in SMC and AECII while none of the controls showed DT staining (**A**). All cell types (SMC, AECII and EC were stained for *NOX4* in IPF patients, while only EC were stained in the control (**B**). Macrophages were stained with both antibodies in both control and IPF (most likely not specific). SMC: smooth muscle cells, EC: endothelial cells, AECII: alveolar epithelial cells type 2, MAC macrophages.

**Table 1 antioxidants-10-01833-t001:** RT-qPCR primers used in the study.

Gene	Accession Number	Primers Forward 5′–3′	Primers Reverse 5′–3′
*NOX4*	NM_001143836.3	AACCGAACCAGCTCTCAGAA	TTGACCATTCGGATTTCCAT
*ACTA2*	NM_001141945.2	AATACTCTGTCTGGATCGGTGGCT	ACGAGTCAGAGCTTTGGCTAGGAA
*FN1*	NM_001306129.2	CGGTGGCTGTCAGTCAAAG	AAACCTCGGCTTCCTCCATAA
*COL1A1*	NM_000088.4	GAGGGCCAAGACGAAGACATC	CAGATCACGTCATCGCACAAC
*GAPDH*	NM_001256799.3	GCACAAGAGGAAGAGAGAGACC	AGGGGAGATTCAGTGTGGTG
*B2M*	NM_004048.4	TGCTCGCGCTACTCTCTCTTT	TCTGCTGGATGACGTGAGTAAAC

**Table 2 antioxidants-10-01833-t002:** Additional information of human lung tissue samples used in the study.

Patient Number	Pathology	Age at Diagnosis	Gender	Origin of Sample	Lung Region
1	IPF	77	M	Surgical biopsy	Right upper lobe
2	IPF (fibrotic hypersensitivity pneumoniae)	69	M	Surgical biopsy	Right upper lobe
3	IPF	74	M	Surgical biopsy	Left lower lobe
4	IPF	71	M	Surgical biopsy	Left lower lobe
5	IPF (pleural fibrosis)	75	N/A	Surgical biopsy	N/A
6	Papillary adenocarcinoma	62	M	Biopsy, adjacent tissue	Right middle lobe
7	Undifferentiated lung adenocarcinoma	69	M	Biopsy, adjacent tissue	Left upper lobe
8	Metastases from endometrial adenocarcinoma	56	F	Biopsy, adjacent tissue	Left lower lobe
9	Bullous emphysema	30	M	Biopsy, adjacent tissue	Right upper lobe
10	Acute bronchitis	85	M	Postmortem material	N/A

IPF: idiopathic pulmonary fibrosis; M: male; F: female; N/A: not available.

## Data Availability

Data is contained within the article or Appendix A.

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
