# Peer review of "Di-Tyrosine Crosslinking and NOX4 Expression as Oxidative Pathological Markers in the Lungs of Patients with Idiopathic Pulmonary Fibrosis"

_antioxidants, 2021, doi:10.3390/antiox10111833_

Round 1

Reviewer 1 Report

I really enjoyed reading the manuscript by Blaskovic et al. The subject is really interesting and the results are important for future studies aiming at clarifying the roles of NOX4-derived ROS and oxidative protein modification dityrosine in idiopathic pulmonary fibrosis (IPF) pathology. NOX4 did not conclusively correlate with dityrosine in IPF samples. Nevertheless, the specific and validated antibody facilitated establishment of dityrosine as a novel histopathological marker of IPF pathology.

The writing style and the use of language were excellent.

I recommend publishing the manuscript with minor modifications; more info on the human lung samples could be added into Table 2 or text. Which samples are postmortem? Are cancer patient samples from the tumor or healthy lung tissue? How many samples per patient were stained and from which parts of lungs? What is known about NOX4 and DT in lung cancer, bronchitis and bullous emphysema? Should there be more NOX4 staining in these conditions compared to healthy lung tissue?

Author Response

Reviewer 1.

Comments and Suggestions for Authors

I really enjoyed reading the manuscript by Blaskovic et al. The subject is really interesting and the results are important for future studies aiming at clarifying the roles of NOX4-derived ROS and oxidative protein modification dityrosine in idiopathic pulmonary fibrosis (IPF) pathology. NOX4 did not conclusively correlate with dityrosine in IPF samples. Nevertheless, the specific and validated antibody facilitated establishment of dityrosine as a novel histopathological marker of IPF pathology. The writing style and the use of language were excellent.

We thank reviewer 1 for this very positive feedback and for reading carefully the manuscript. Please find below a point by point answer to all the questions raised by reviewer 1:

I recommend publishing the manuscript with minor modifications;

  1. More info on the human lung samples could be added into Table 2 or text.

A1. Table 2 was modified to include the origin of the samples

  1. Which samples are postmortem?

A2. All samples were surgical biopsies except nr 10, who was postmortem (see updated table 2 below).

  1. Are cancer patient samples from the tumor or healthy lung tissue?

A3. Adjacent healthy tissue was used in control samples. The following sentence was added in the text: “At least 10 adjacent sections were stained for each antibody. Control tissue consisted of adjacent healthy tissue for patients nr 6-9 and from post-mortem tissue for patient nr 10”.

  1. How many samples per patient were stained and from which parts of lungs?

A4. One sample per patient, 10 slices of  each sample were stained for each patient: The regions of the lungs are now documented in Table 2 and the above sentence was added in the text.

  1. What is known about NOX4 and DT in lung cancer, bronchitis and bullous emphysema? Should there be more NOX4 staining in these conditions compared to healthy lung tissue?

A5. According to protein atlas data NOX4 is not a prognostic marker in lung cancer. Ciliary dysfunction is increased in neutrophilic asthma and associated with increased NOX4 expression (https://pubmed.ncbi.nlm.nih.gov/26836936/). Also, expression of NOX4 in Smooth Muscles of Small Airway Correlates with the Disease Severity of COPD (https://www.ncbi.nlm.nih.gov/pmc/articles/PMC5021463/). In terms of DT formation, one study showed increased DT in a model of COPD induced by COPD https://pubmed.ncbi.nlm.nih.gov/21997333/

Thus, one would expect an increased NOX4 expression and DT formation in some but not all pulmonary pathologies. However, in our study, we used adjacent “healthy” tissue and did not observe strong induction of DT.

Reviewer 2 Report

Minor comments:

  1. It will be great use lung fibroblasts for in vitro
  2. Control lung sample information is missing.
  3. Figure 1A and Suppl Figure 1A: qPCR data analysis is not clearly described. Each gene expression is normalized by GAPDH and B2M both or either of two genes? All of the gene name should be capitalized. For example, fibronectin gene should be FN1, Col1A should be: COL1A1 or COL1A2  If Figure 1A is qPCR result.
  4. Figure 6 results: How defined NOX4 and DT positive cell type ? If there is no double stain it is hard to consider each cell type just by morphology.

Author Response

Reviewer 2.

Comments and Suggestions for Authors

We thank reviewer 2 for his comments and suggestions, which helped us improving the manuscript. Please find below a point by point answer to all questions raised by reviewer 2.

Minor comments:

  1. It will be great use lung fibroblasts for in vitro

A1. The MRC-5 cells described on lane 107 and used in Fig.1A,B and Suppl fig.1 are lung fibroblasts isolated from a 14-week old aborted Caucasian male foetus, see https://www.atcc.org/products/ccl-171

  1. Control lung sample information is missing.

A2. Table 2 was modified to include additional information about the sex, the age of controls and the origin of all samples (see updated table 2 below).

  1. Figure 1A and Suppl Figure 1A: qPCR data analysis is not clearly described. Each gene expression is normalized by GAPDH and B2M both or either of two genes?

A3. qPCR data were normalized for both GAPDH and B2M, according to the method described here: Vandesompele J, De Preter K, Pattyn F, Poppe B, Van Roy N, De Paepe A, Speleman F. Accurate normalization of real-time quantitative RT-PCR data by geometric averaging of multiple internal control genes. Genome Biol. 2002 Jun

  1. All of the gene name should be capitalized. For example, fibronectin gene should be FN1, Col1A should be: COL1A1 or COL1A2  If Figure 1A is qPCR result.

A4. Thank you for your comment. The labelling of qPCR data was modified accordingly.

  1. Figure 6 results: How defined NOX4 and DT positive cell type? If there is no double stain it is hard to consider each cell type just by morphology.

A5. This is a valid point, and for this reason we did not characterize the alveolar epithelial type I cells, because it is impossible to properly identify them without specific cell type markers. For the other cell types, we consulted a lung pathologist (J.C. Pache): endothelial cells, smooth muscle cells, alveolar epithelial cells type II and macrophages are recognized by their tissue localization and morphology. No double staining is required for these cells.A sentence was added on lane 276 to address this point.

Table 2

Patient number

Pathology

Age at diagnosis

Gender

Origin of sample

Lung region

1

IPF

77

M

Surgical biopsy

Right upper lobe

2

IPF (fibrotic hypersensitivity pneumoniae)

69

M

Surgical biopsy

Right upper lobe

3

IPF

74

M

Surgical biopsy

Left lower lobe

4

IPF

71

M

Surgical biopsy

Left lower lobe

5

IPF (pleural fibrosis)

75

N/A

Surgical biopsy

N/A

6

Papillary adenocarcinoma

62

M

Biopsy, adjacent tissue

Right middle lobe

7

Undifferentiated lung adenocarcinoma

69

M

Biopsy, adjacent tissue

Left upper lobe

8

Metastases from endometrial adenocarcinoma

56

F

Biopsy, adjacent tissue

Left lower lobe

9

Bullous emphysema

30

M

Biopsy, adjacent tissue

Right upper lobe

10

Acute Bronchitis

85

M

Post-mortem material

N/A